# Long Non-Coding RNA Expression Levels Modulate Cell-Type-Specific Splicing Patterns by Altering Their Interaction Landscape with RNA-Binding Proteins

**DOI:** 10.3390/genes10080593

**Published:** 2019-08-06

**Authors:** Felipe Wendt Porto, Swapna Vidhur Daulatabad, Sarath Chandra Janga

**Affiliations:** 1Department of BioHealth Informatics, School of Informatics and Computing, IUPUI, Indianapolis, IN 46202, USA; 2Department of Medical and Molecular Genetics, Indiana University School of Medicine, Indianapolis, IN 46202, USA; 3Centre for Computational Biology and Bioinformatics, Indiana University School of Medicine, Indianapolis, IN 46202, USA

**Keywords:** long non-coding RNA, cell-type-specific, alternative splicing, functional enrichment, RNA-binding proteins, protein binding lncRNA sponges, secondary RNA structure, cancer

## Abstract

Recent developments in our understanding of the interactions between long non-coding RNAs (lncRNAs) and cellular components have improved treatment approaches for various human diseases including cancer, vascular diseases, and neurological diseases. Although investigation of specific lncRNAs revealed their role in the metabolism of cellular RNA, our understanding of their contribution to post-transcriptional regulation is relatively limited. In this study, we explore the role of lncRNAs in modulating alternative splicing and their impact on downstream protein–RNA interaction networks. Analysis of alternative splicing events across 39 lncRNA knockdown and wildtype RNA-sequencing datasets from three human cell lines—HeLa (cervical cancer), K562 (myeloid leukemia), and U87 (glioblastoma)—resulted in the high-confidence (false discovery rate (fdr) < 0.01) identification of 11,630 skipped exon events and 5895 retained intron events, implicating 759 genes to be impacted at the post-transcriptional level due to the loss of lncRNAs. We observed that a majority of the alternatively spliced genes in a lncRNA knockdown were specific to the cell type. In tandem, the functions annotated to the genes affected by alternative splicing across each lncRNA knockdown also displayed cell-type specificity. To understand the mechanism behind this cell-type-specific alternative splicing pattern, we analyzed RNA-binding protein (RBP)–RNA interaction profiles across the spliced regions in order to observe cell-type-specific alternative splice event RBP binding preference. Despite limited RBP binding data across cell lines, alternatively spliced events detected in lncRNA perturbation experiments were associated with RBPs binding in proximal intron–exon junctions in a cell-type-specific manner. The cellular functions affected by alternative splicing were also affected in a cell-type-specific manner. Based on the RBP binding profiles in HeLa and K562 cells, we hypothesize that several lncRNAs are likely to exhibit a sponge effect in disease contexts, resulting in the functional disruption of RBPs and their downstream functions. We propose that such lncRNA sponges can extensively rewire post-transcriptional gene regulatory networks by altering the protein–RNA interaction landscape in a cell-type-specific manner.

## 1. Introduction

One of the major challenges in the post-genomic era is to understand fundamental mechanisms of long non-coding RNAs (lncRNAs) and their role in modulating cellular homeostasis. Despite increased awareness of their influence on alternative splicing and alterations in notable cancers, most lncRNAs are still not known to have a function and thousands of lncRNAs are believed to be a result of transcriptional noise [1]. Their length and low expression have acted as a barrier for experimental assays and for building computational models [2,3], resulting in a lack of approaches to confidently assess the full scope of lncRNA function and structure. However, with the help of novel genome editing technologies such as “catalytic dead” Cas9 (dCas9) silencing [4], genome-scale probing methods which study the loss of function phenotypes for lncRNAs are beginning to emerge.

By combining sequential dCas9-KRAB guides predicted via in silico models, effective knockdowns of whole lncRNA genes have recently been made, providing promise for expanding genome-wide knockdown protocols [4]. With the help of such improved lncRNA knockdown experiments, one can begin to confidently analyze and associate the downstream phenotypic changes and resulting interaction networks of lncRNAs across cell types. CRISPR (Clustered Regularly Interspaced Short Palindromic Repeats)-guided knockdown assays [5] and large-scale lncRNA functional analysis [6] demonstrate the translational significance of CRISPR-mediated lncRNA perturbations.

Other studies of lncRNAs have repeatedly shown that lncRNAs perturb or enhance biological functions in a cell-type- or tissue-specific manner [7,8]. Putative lncRNAs like HOTAIR, NEAT1, and MALAT1 have been demonstrated to interact with specific RNA-binding proteins (RBPs) like HUR and ELAVL1 in a tissue-specific fashion [8,9], with increasing evidence for the broader role of lncRNAs in modulating a variety of post-transcriptional processes [10,11]. However, few databases have curated useful cross-linking immunoprecipitation (CLIP) [12] data with lncRNAs and annotated functional effects that match the quality of Seten, [13] CLIPdb’s POSTAR2 [14], and ENCODE [15]. RBPs have been shown to influence alternative splicing and their interaction with pre-mRNAs [16], thereby impacting cellular functions and phenotypes. Yet, in a post-transcriptional regulatory context, our understanding of the association between RBPs and the non-coding transcriptome remains unclear.

To understand the influence of lncRNAs on alternative splicing, we integrated dCAS9-mediated lncRNA knockdown RNA-sequencing data [4] across three cancer cell lines along with protein–RNA interaction maps from eCLIP [17] experiments performed as part of the ENCODE project. This study examines the role of lncRNAs in modulating the cell-type-specific alternative splicing outcomes due to their physical interactions with RBPs.

## 2. Materials and Methods

To dissect the functional impact of lncRNAs on splicing, we downloaded publicly available RNA-Seq data from a study (GSE85011) [4] where multiple lncRNAs were knocked down across three human cancer cell lines (Table 1). As illustrated in the workflow (Figure 1), the RNA-Seq data were aligned to the human reference genome. The aligned data were further analyzed to identify alternative splicing events across lncRNA knockdowns and corresponding control samples. RBP binding profiles were analyzed to understand their binding preference around alternative splice events. The following sections discuss individual steps involved in our workflow in detail.

### 2.1. Sequence Alignment of RNA-Seq Reads Using HISAT

Data from the study GSE85011 were generated by creating a guide RNA library to facilitate CRISPRi (CRISPR interference) lncRNA knockdowns via precise heterochromatization [4]. Single-end RNA-Seq data of lncRNA knockdowns and controls in the HeLa (cervical cancer), U87 (glioblastoma), and K562 (leukemia) cell lines were downloaded from GEO [18] in the “fastq” format using NCBI’s sratoolkit (version 2.8.2) [19]. We used a highly efficient short-read RNA-Seq alignment tool, HISAT2 [20], to align the downloaded data to the human reference genome. Default parameters of HISAT2 were used to align the reads against the hg38 reference genome. By utilizing SAMtools (version 1.3.1) [21] for data processing, the SAM (Sequence Alignment/Map) files from HISAT2 alignment were converted to the BAM (Binary Alignment/Map) format. Concurrently, the BAM files were transformed into the sorted-BAM format. The average alignment rate across the samples was 91%. The corresponding SRA (Sequence Read Archive) sample run IDs (SRR IDs) and their respective alignment statistics are reported in Table 1.

### 2.2. Identifying Differentially Spliced Events Using Replicate Multivariate Analysis of Transcript Splicing (rMATS)

RNA-Seq data enable us to observe alternative splicing events across two given conditions. In this study we deployed rMATS (version 3.2.5) [22] on samples corresponding to 39 lncRNA knockdowns and their respective controls to identify the differential alternative splice (AS) events. rMATS takes sorted-BAM files as input and identifies a variety of splicing events that are differential across two conditions. We used sorted-BAM files of lncRNA knockdowns, control reads, and corresponding replicates as input for rMATS. Additionally, human transcript annotation was provided with a GTF (Gene transfer format, version 84) file from Ensembl [23]. By implementing our inputs into the pipeline with default thresholds, we could compare AS patterns in the presence and absence of a lncRNA. Thereby, rMATS enabled us to analyze the inclusion/exclusion of target exons/introns pertaining to different types of AS events: skipped exon (SE), alternative 5′ splice site (A5SS), alternative 3′ splice site (A3SS), mutually exclusive exons (MXE), and retained intron (RI) across lncRNA knockdown experiments. We ran rMATS on 54 comparisons of samples pertaining to 39 lncRNA knockdowns in three cell lines; however, not all the lncRNAs had knocked down RNA-Seq data available across the three cell lines.

Custom Python scripts were developed to parse through rMATS output files across all comparisons to generate AS event frequency matrices. A filter of false discovery rate (fdr) <10 and *p*-value < 0.05 was employed over these tables, thereby enabling us to extract only the significant AS events (Appendix A).

### 2.3. Visualizing Incidence of Differentially Spliced Genes in lncRNA Knocked Down Cell Lines

The frequency matrices generated from the rMATS output were further analyzed to gauge how often each gene was alternatively spliced when a specific lncRNA was knocked down in a given cell line. To optimize data processing and visualization, the matrix was further filtered to remove instances which were not populated with data. The genome-wide influence of lncRNAs on splicing was visualized by generating a heatmap depicting the incidence of a gene being alternatively spliced in a lncRNA knockdown using Morpheus [24], a heatmap generation tool from Broad Institute. Both rows and columns were clustered hierarchically based on the “one minus Spearman rank correlation” distance option. The Spearman distance option was the most relevant method to discriminate the genes’ cell-type specificity, as it is a non-parametric test and gave the most consistent groupings after trials of similar distance metrics. We used Venny (v2.1) [25] to generate Venn diagrams to depict the differentially spliced gene overlaps across cell lines.

### 2.4. Functional Enrichment Analysis of Alternatively Spliced Genes

To understand the functional impact of lncRNAs on cellular functions, we performed functional enrichment analysis using ClueGO (v2.50) [26] over alternatively spliced gene sets from each lncRNA knockdown. ClueGO is a Cytoscape [27] plug-in which enables users to perform comprehensive functional interpretation of gene sets. ClueGO can analyze gene sets and visualize the respective functionally grouped annotations based on pathway enrichments recognized in the corresponding gene sets. We extracted the gene sets that were exclusively alternatively spliced in a given cell line and plugged them into ClueGO, thereby extracting functional contributions of lncRNAs. We used all the Gene Ontology (GO) terms irrespective of their level in the GO hierarchy for identifying enrichment terms after searching for term annotations from databases like KEGG [28], GO Cellular Components [29,30], GO Biological Process [29,30], and REACTOME Pathways [31]. Only significantly enriched terms (*p*-value < 0.05) were used for the downstream analysis. The functional enrichment of the smaller gene sets affected by alternative splicing across cell types revealed no significant (*p*-value > 0.05) pathways or functional annotations.

### 2.5. Extraction of RBP Binding Profiles across AS Events

Experimental strategies like CLIP [12], iCLIP [32], and eCLIP [17] have significantly improved our understanding of the RNA bound proteome. By observing variations of RBP binding preferences around AS events in the presence and absence of a given lncRNA, we gained a new perspective on the functional dynamics and crosstalk between lncRNAs and RNA-binding proteins (RBPs). Publicly available eCLIP experimental data in the HeLa and K562 cell lines were obtained from the ENCODE project [15]. The eCLIP data were then processed to extract the genome-level RBP binding profiles in both cell lines.

Additionally, we extracted the genomic loci where the AS events occurred from the rMATS output data. Potential genomic loci involved in AS can be captured by examining adjacent upstream and downstream regions in the exon start and end loci. A total of six types of proximal coordinates were extracted: upstream exon start and end, AS event-specific exon start and end, and the downstream exon start and end. Each of the six coordinates were given a distance allowance of 500 base pairs on either side of the sites to confirm locations which overlapped with the binding sites of RBPs. Both RBP binding profiles and AS event sites were processed and converted into the BED (Browser Extensible Data) format. Bedtools (version 2.18.1) [33] is one of the most efficient tools to analyze genomic loci-based information. We used the Bedtools-intersect option to extract the overlaps across RBP binding sites and AS events and proximal loci in order to understand which RBPs are potentially associated with a specific AS event [see Appendix A]. However, the eCLIP data on ENCODE did not host high-confidence U87 cell line experiments, therefore no RBP binding profiles from U87 were extracted.

### 2.6. Enrichment of RBP Binding over Alternative Splicing Events

Although we identified RBP and AS event interactions based on eCLIP data, we wanted to identify only the most likely interactions for further analysis. A hypergeometric statistical test was performed to observe the enrichment of an RBP’s binding preference with respect to each lncRNA that was knocked down, thereby determining significant relative RBP binding preference.

The relationship between RBP binding frequencies was highly influenced by many missing values, which might not be representative of the whole relationship of binding. The importance of a reported binding had no numerical value attached, so a hypergeometric probability was needed. A majority of the lncRNAs had a higher proportion of binding rather than non-binding, and Fisher’s exact odds ratio test evaluated the representation of binding. A Fisher’s exact test was run for each of the bound and unbound genes to a motif created by a lncRNA knockdown and all the genes from each lncRNA. The *p*-values from each Fisher’s test were then adjusted in reference to all the other Fisher’s exact tests.

Table 2 shows the contingency table of bound and unbound protein frequencies used to conduct the Fisher’s exact test. Each square in the heatmaps shown in Figure 4A,B demonstrate the significance (-log(corrected *p*-value)) from each of these tests. The maps were clustered hierarchically to see if there was a relationship between RBP and lncRNAs.

### 2.7. Correlation Analysis of lncRNA Specific RBP Binding Activity and AS Events

A correlation analysis was performed across the number of skipped events with the amount of RBPs relative and each lncRNA specifically to determine a relationship between RBP binding frequency and alternative splicing events. However, a significant correlation was not observed given the small amount of data points. Appendix A depicts the attempts and tables used to show the connection between the lncRNAs and RBP.

## 3. Results

### 3.1. Framework for Studying Alternative Splicing Outcomes in lncRNA Knockdown RNA-Sequencing Experiments

In this study, we investigated alternative splicing events, functions of alternatively spliced gene sets, and RBP-lncRNA binding patterns across 39 lncRNA knockdowns in HeLa (cervical cancer), K562 (myeloid leukemia), and U87 (glioblastoma) cell lines. An overview of the three-step analysis is illustrated in Figure 1 (see Materials and Methods). First, we collected and processed the raw RNA-Seq data for multiple lncRNA knockdown samples in human cancer cell lines and identified splicing events using the replicate multivariate analysis of transcript splicing (rMATS) [22] pipeline. Secondly, alternative splicing summaries were analyzed, and functional enrichment analysis of the effected gene sets was performed. Third, we analyzed the binding profiles of RNA-binding proteins (RBPs) with respect to the skipped exon (SE) and retained intron (RI) events across 39 lncRNA knockdowns, finding that the RBPs’ binding profiles around SE and RI events were also unique to each cell line. Additionally, we observed RBPs’ substantial binding preference for knocked down lncRNAs and proposed that certain lncRNAs demonstrate a sponge-like RBP binding activity.

### 3.2. Skipped Exon Events Followed by Retained Intron Events Are the Most Prominent Alternative Splicing Events Occurring Due to the Loss of lncRNAs across Multiple Human Cell Lines

We collected the RNA sequencing data from dCAS9-based lncRNA knockdown study [4] which contained 39 lncRNA knockdowns in U87, HeLa, and K562 cell lines (see Materials and Methods). LncRNA knockdown sequence replicates and control samples were aligned onto the human reference genome (hg38) using HISAT2 [20]. The overall percentage of alignment is highlighted in Appendix A, demonstrating a high fraction of read alignment to the reference genome (average alignment rate ≥ 91%). The alternative splicing (AS) event identification analysis was performed by deploying the rMATS pipeline over the three cell lines. A filter of fdr < 0.1 and *p*-value < 0.05 was employed to extract the most likely alternative splicing events (see Materials and Methods). A total of 26,167 high-confidence AS events were extracted from the rMATS analysis output across three cell lines. Skipped exon (SE) events were the most predominant events, constituting 44.4% of the total events, followed by retained intron (RI) events at 22.5%. Sample-wise distributions of splice events are provided as Appendix A. The output from the rMATS analysis was parsed and summarized into a total of 17,525 unique, statistically significant splice events (11,630 SE and 5895 RI) and their AS event frequency across 39 lncRNA knockdowns in the three cell lines is available as Appendix A.

### 3.3. Hierarchically Clustered Heatmaps of AS Event Frequency in lncRNA Knockdowns Reveal Cell-Type-Specific Alternative Splicing Signatures

Annotated splicing summaries were compiled to generate matrices depicting the frequency of alternative splicing events occurring in lncRNA knockdowns across three different cell lines. In order to visualize the AS events from the matrix, heatmaps were generated for SE and RI events. A legible resolution of the SE and RI heatmaps was obtained after missing data were filtered out. Across RI and SE events, we observed that the alternatively spliced gene clusters were predominantly unique across each cell line (Appendix A). Consequently, even though the same lncRNA was knocked out across two cell lines, the genes that were alternatively spliced were cell-type specific. SE events had the most pronounced cell-type specificity, in comparison to other AS events. In tandem, we also observed that the cellular and phenotypic functions affected by genes alternatively spliced across SE and RI events were specific to each cell line.

Among the lncRNA knockdown dataset described in the Materials and Methods, we found only 12 lncRNAs to be common across three cell lines. Amongst all the AS events in the three different cell lines, no single genes were affected in the same way by a knockdown. The resulting alternatively spliced genes from the 12 lncRNA knockdowns which were present in at least two cell types were observed to be cell-type specific. Figure 2 illustrates how the cell lines which had a lncRNA knocked down affected the frequency of SE events in specific genes. The genes affected by RI events were also altered in a cell-type-specific manner. However, the intensity of the signal was relatively lower in RI events when compared to SE events.

In the AS event frequency heatmap, the SE events had very defined clusters and continued to be clustered in a distinct cell-type-specific manner [see Appendix A]. The genes which were annotated with SE events were rarely shared across two cell lines. Out of the total 759 alternatively spliced genes, only 7.3% (37 genes) were alternatively spliced across at least two cell lines. When enrichment analysis was performed over the few SE affected genes which were shared across different cell lines, there was too little information for a statistically significant functional annotation to emerge. The number of SE-event-affected genes shared across cell lines can be seen in the Venn diagram in Figure 2. Similarly, a heatmap of RI event frequencies was generated. Akin to the observations from SE event analysis, the gene sets with RI events were mostly cell-type specific [see Appendix A].

The gene sets affected by RI events were cell-type specific, but these gene sets overlapped slightly more among the cell lines, relative to SE distributions. Given the small number of genes alternatively spliced and RI events shared (maximum of 30 shared events), the scope for significant functional annotation was low. The 162 genes affected by RI events in the K562 cell line had a distinct signal from the other cell lines, further emphasizing their cell-type specificity.

### 3.4. lncRNA Knockdowns Not only Enriched for Cell-Type-Specific Functions but Occasionally Favor Similar Functions via Varied Alternatively Spliced Gene Sets

In an attempt to understand the downstream functions being influenced by lncRNAs, functional enrichment analysis was performed over 759 alternatively spliced genes corresponding to 39 lncRNA knockdowns in three cell lines (see Materials and Methods). Only 28 genes were shared between the SE and RI events, while 481 genes were unique to SE events and 250 genes were unique to RI events. Figure 3 illustrates the significant (*p* < 0.05) potential functions associated to genes alternatively spliced in each cell line resulting from a lncRNA knockdown. As expected, the functions affected within each cell line were also unique to each cell line. On the rare occasion of a common enriched function between cell lines, gene set composition leading to that function varied drastically.

#### 3.4.1. Genes Alternatively Spliced via SE Events in U87, HeLa, and K562 Reveal Cell-Type-Specific Functional Enrichment

SE events affected a total of 509 genes, and the respective functional annotation terms were enriched from four different pathway databases (see Materials and Methods). A unique gene set of 138 genes corresponding to 14 terms were enriched in the U87 cell line. The highest number of genes that were alternatively spliced in samples were induced by the knockdown of lncRNA families XLOC and RP11. The U87 cell-type-specific gene cluster had areas of higher magnitude of splicing events and shared only 25 genes with the K562 cell line. Of the 25 shared genes no significant pathways were annotated. The top functions affected by the alternative splicing induced by the lncRNA knockdowns in the U87 cell line correspond with DNA endonuclease repair (37% of genes) and ribosomal complex formation (16% of genes) [see Appendix A]. These functional observations are in coherence with the literature, as it is known that endonuclease repair activity is a function targeted by cancerous U87 [34].

The 141 genes affected in the K562 cell line were enriched for a total of 17 functional terms. The top functions affected by SE events in the K652 cell line were cell microfiber construction (45% of genes), phosphotransferase activity (23% of genes), and p53 signaling pathway (12% of genes) [see Appendix A]. Functional annotations like cell growth and cell death have been identified as key checkpoints in various cancers, including leukemia (K562) [35]. However, this study appears to provide a novel functional annotation of “cell microfiber construction” being affected in leukemia (K562). Thereby, our genome-wide functional analysis is not only able to predict novel functions based on genes affected by AS events, but also support functions previously annotated to these cell lines.

The HeLa clusters affected genes in a large gene cluster, with 193 unique genes over-represented with 28 functional terms. The top functions affected by SE events in the HeLa cell line were deoxyribonucleoside monophosphate metabolic processes (29% of genes) and vesicle formation and vesicle movement (15% of genes) [see Appendix A]. Vesicle formation and movement were observed to be attenuated in the HeLa cell line in a gene (SPIN90) dependent manner, which our analysis was able to capture [36]. No significant pathways were annotated for the 12 genes the HeLa cell line shared with the K562 cell line.

#### 3.4.2. Genes Alternatively Spliced via RI Events in HeLa and K562 Reveal Cell-Type-Specific Functional Enrichment

RI events affected a total of 278 genes across the three cell lines. The respective functional annotation terms were enriched from four different pathway databases (see Materials and Methods). While the clustered gene set in the HeLa cell line yielded no significant (*p* < 0.05) pathways, the knockdown of RP5-1148A21.3 in the HeLa cell line was able to alternatively splice all 278 genes [see Appendix A]. Thus, the 278 genes affected in the HeLa cell line by RP5-1148A21.3 enriched 53 functional terms: responses to endoplasmic reticulum stress (19% of genes), holiday junction resolvase complex (9% of genes), and RNA splicing (8% of genes). The few genes affected within the U87 cell line did not enrich any significant pathways.

The 162 genes affected in the K562 cell line were enriched for 18 functional terms composed of concepts such as “ribosomal construction” and “negative regulation of autophagy”. “Cytosolic large ribosomal subunit” constituted 50% of the genes alternatively spliced in the K562 gene set [see Appendix A]. Other prominent functions associated with K562 were ribosomal RNA processing in the nucleus and cytosol (7% of genes), and negative regulation of autophagy. The functional annotation of the term “targeted rRNA processing” onto K562 is in coherence with the literature, as “targeted rRNA processing” has been identified to play a key role in K562 cells [37].

### 3.5. Analysis of lncRNA AS Events Proximal to RBP Binding Sites Reveals Cell-Type-Specific Interactions and Supports a lncRNA–RBP Sponge Model

As a conduit to understand lncRNAs’ role in alternative splicing, lncRNAs’ interactions with RBPs were extracted. RBP binding profiles for 22 lncRNAs were obtained from documented eCLIP experiments from the ENCODE database. Appendix A highlights the binding profiles which overlapped with proximal (±500 bp) alternative splice site locations, revealing 4,261,897 RBP binding locations for 148 RBPs on 22 lncRNAs. The significance of relative frequencies of bound and unbound RBP sites was gauged by deploying Fisher’s exact test on each lncRNA’s RBP binding preferences. Figure 4 showcases the intensity of each RBP’s interactions to their respective 11 lncRNAs in the K562 cell line (Figure 4A) and 14 lncRNAs in the HeLa cell line (Figure 4B). Additionally, we observed lncRNAs which acted as RBP sequestering sponges, which is illustrated in Figure 4C, based on their extensive interactions with RBPs. Figure 4D demonstrates how sponging lncRNAs like LINC00909 have many interactions with a variety of RBPs.

The lncRNA–RBP binding-profile-based clustering analysis across both cell lines was not very informative. However, an interesting behavior was revealed where certain lncRNAs (e.g., LINC00909, LINC00263, and LINC00910) had many binding events across many RBPs. Therefore, the downstream alternative splicing caused by the loss of a lncRNA is induced by the absence of an RBP binding sponge. As highlighted in the model shown in Figure 4C, in cancer cells a lncRNA might bind to many RBPs, where its expression level could facilitate extensive RBP interactions. However, in the event of a lncRNA knockdown or due to the loss of function of a lncRNA, an abundance of RBPs interact with pre-mRNA targets illustrated in Figure 4C, thereby inducing alternative splicing.

As highlighted in Figure 4A, most lncRNAs in the K562 cell line bound generally to RBPs like KHSRP, CSTF2T, YBX3, ZNF622, SAFB2, SRSF1, and QKI. LncRNAs LINC00910, LINC00680, RP11-392P7.6, and LINC00909 showed a very high number of RBP interactions in the K562 cell line (see Appendix A, Figure 4D) and exemplify the proposed RBP sponge binding model. Other interesting patterns of lncRNA–RBP binding included distinct RBP binding preferences for lncRNAs from the same family, namely XLOC_042889 and XLOC_038702.

Despite only having 10 RBPs binding within the HeLa cell line proximal AS events, lncRNA LINC00909′s RBP interactions further reinforced our proposed lncRNA–RBP sponge model. As illustrated in Figure 4B, RBPs ELAVL1 and HNRNPU were observed to have many null values across lncRNA knockdown samples and one RBP (HNRNPC) had many significant binding associations across all lncRNA knockdowns. The parameters of Fisher’s exact test require bound and unbound frequencies of the genes, which can be observed in the contingency table (Table 1), and thus any RBP which is not reported to be unbound will yield a null result. Both ELAVL1 and HNRNPU were manually checked across all lncRNAs in the HeLa cell line, and they only had values for binding across lncRNAs. Thus, the binding preferences of ELAV1 and HNRNPU were very indifferent and were not considered as contributors to the RNA-binding protein sponge model; however, the binding specificity of these RBPs could be revealed as more interaction data are collected across other lncRNAs.

## 4. Discussion

In this study, we investigated the splicing alterations in lncRNA knockdown experiments, and depicted a molecular mechanism of RBPs’ influence on alternative splicing. The analysis of the alternative splicing heat maps showed that transcriptional networks were perturbed in a cell-type-specific pattern. The observed cell-type-specific pattern of perturbation is in line with previous profiling of expression in lncRNA knockdowns [4]. Unique to this experiment, alternative splicing displayed characteristics of being cell-type specific over many lncRNAs, with extreme examples depicted in the SE experiments.

Based on our RBP binding analysis, lncRNAs display RBP sponge-like behavior and hint at other methods of inducing alternative splicing. This study also attempted to implicate that lncRNAs like LINC00909 and LINC00910 act as sponges; however, there could be other means of inducing splicing. For instance, lncRNAs like RP5-1148A21.3 seem to participate in a different way, which is depicted in its high involvement in RI events, with only 18 RBPs reported to be bound. Other studies implicate lncRNAs’ potential interaction with themselves to make silencing structures, crystalline structures, or molecular machinery [9,38]. Additionally, subnuclear bodies have shown promise in understanding lncRNAs’ potential in creating other cellular machinery [39], and lncRNAs which bind to many RBPs could be recruiting those proteins to form a subnuclear body. Groups invested in mechanisms of post-transcriptional regulation may begin to examine RBP binding data and multiple sequence alignment of lncRNAs in order to understand intramolecular interactions in subnuclear bodies and secondary lncRNA structure [40,41]. Hence, further attempts to understand non-coding RNA structure should account for interactions of lncRNAs and RBPs inside of the nucleus and in the cytoplasm in order to reveal other means of lncRNAs’ involvement in splicing.

While it is known that lncRNAs bind to microRNAs through a binding sponge mechanism [42], lncRNA–RBP binding activity is not often described as a sponge titration. By sponging microRNAs, lncRNAs revealed many significant instances of their influence on post-transcriptional regulation in a variety of cancers [43,44]. Their influence is controlled through mechanisms like the competitive endogenous model (ceRNA) [45], which will ultimately affect gene expression based on the binding activity of microRNAs. Despite established investigations of miRNA and lncRNA interactions, lncRNA–RBP binding activity is not well explored in the context of post-transcriptional regulation. Other non-coding RNAs like circular RNAs have been reported to act as protein-binding sponges [46], and lncRNAs have been reported to act as binding sponges for the Hur protein [47]. However, this study identified many RBPs binding to a handful of lncRNAs which acted as binding sponges. Thereby, a direct relationship between all the alternative splicing and RBP sponge activity is not clear, but further analysis of RBP and lncRNA binding activity could evaluate the strength of the proposed sponge model.

## 5. Conclusions

Alternative splicing induced by lncRNA knockdowns was shown to be cell-type specific within the human cancer cell lines HeLa, K562, and U87. The downstream cellular functions of 759 genes were significantly affected by 11,630 skipped exon events and 5895 retained intron events, specifically within each cell line. LncRNA and RBP interactions were also shown to be cell-type specific. We hypothesize that several lncRNAs like LINC00909, LINC00910, and LINC00263 are likely to titrate RBPs in cancer, resulting in the functional disruption of RBPs and their downstream functions. We propose that such lncRNA sponges can extensively rewire the post-transcriptional gene regulatory networks by altering the protein–RNA interaction landscape in a cell-type-specific manner. Our study is one of the first to implicate lncRNAs as an RBP sponge, and it reveals more diverse RBP sponge activity than previously observed [41].

However, the sponge model was not able to represent all lncRNAs which played a significant role in splicing. LncRNAs like RP5-1148A21.3, which affected splicing very significantly and had few reported RBPs bound, could have their structure researched. Investigations of lncRNA secondary structure and tertiary structure have revealed methods of epigenetic and post-transcriptomic regulation [40,48], and may further implicate lncRNAs’ influence over alternative splicing. Thereby, we propose that lncRNAs like RP5-1148A21.3 may interact intramolecularly and alter post-transcriptional gene regulatory networks.

Reports on the interactions between lncRNAs in unique human cell lines are becoming more prevalent, and curated databases are emerging to improve the quality of lncRNA annotation [6,49]. As more lncRNA knockdown procedures and RBP binding data are released to the public, pipelines like the one conducted in this study can be utilized to investigate the mechanisms of lncRNAs in alternative splicing.

## Figures and Tables

**Figure 1 genes-10-00593-f001:**
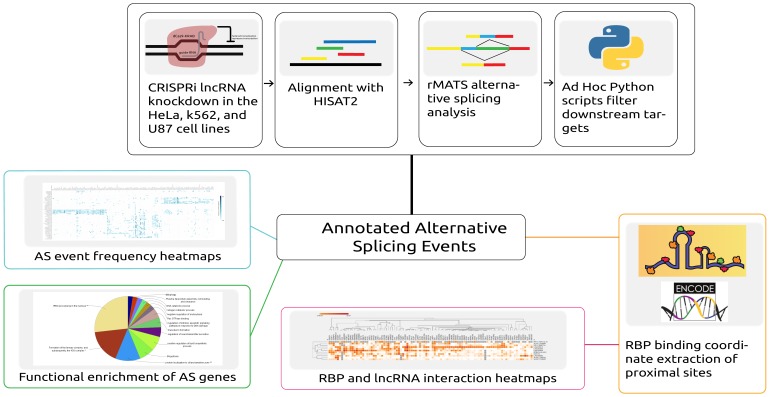
**Depiction of the workflow and analysis conducted in this study.** Automated scripts were developed to download data, align reads to the corresponding reference, and predict the alternative splicing (AS) events across samples. From further processing of the predicted events, both alternative splicing event information and proximal RNA-binding protein (RBP) binding locations were extracted. Alternative splicing events were then analyzed by generating heatmaps and by performing functional enrichment analysis of the alternatively spliced genes. The overlapping proximal RBP binding locations were extracted by analyzing RBP binding profiles obtained from ENCODE.

**Figure 2 genes-10-00593-f002:**
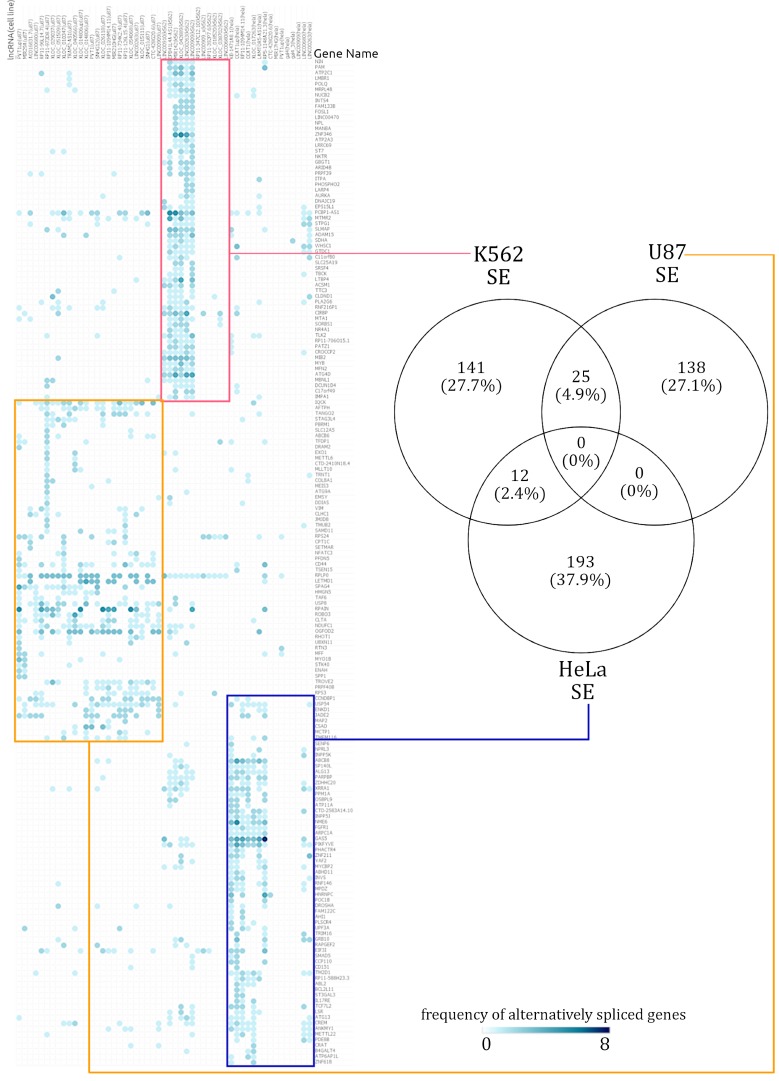
Summary of cell-type-specific skipped exon (SE) events. A hierarchically clustered heatmap demonstrates that the genes spliced via skipping exon isolate themselves in a mostly cell-type-specific manner. The numbers in the Venn diagram show how many alternatively spliced genes are shared across cell lines. Across all the alternatively spliced events in the three different cell lines, no single genes displayed the same pattern.

**Figure 3 genes-10-00593-f003:**
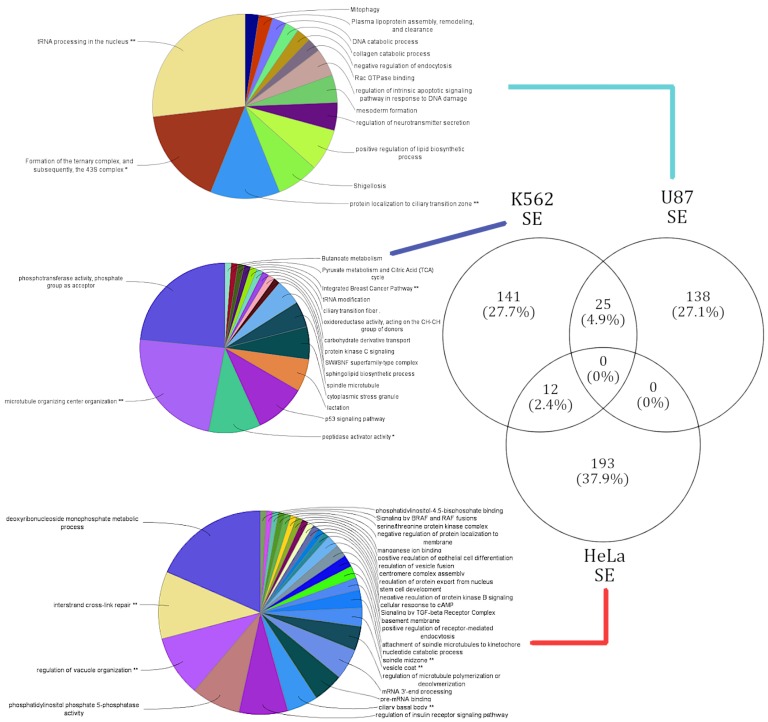
Summary of cell-type-specific GO functional annotation in skipped exon events. Functional annotation of cell-type-specific gene sets that were affected by SE events. The lines connect the corresponding gene sets across each cell type to respective functional annotation. The functions enriched in each cell type are exclusive.

**Figure 4 genes-10-00593-f004:**
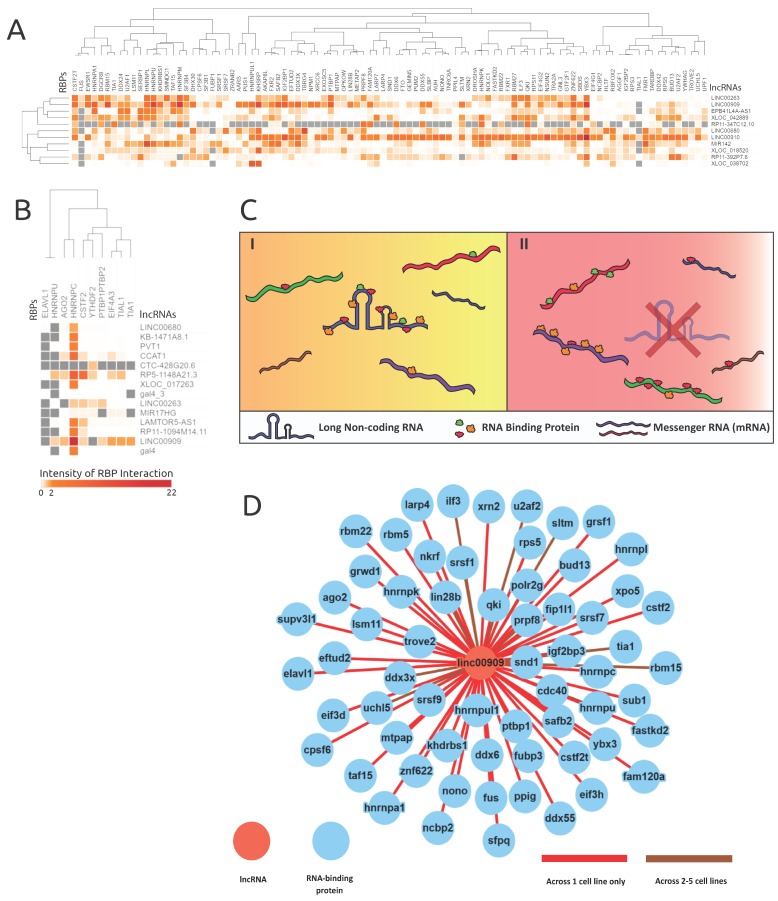
RNA-binding protein eCLIP-based binding profile analyses revealed possible lncRNAs acting as RBP sponges. The intensities are log-scaled false discovery rats (FDRs) of RBP protein interactions with genes alternatively spliced in corresponding lncRNA knockdowns. (**A**) and (**B**) showcase the lncRNAs knocked down in respective cell lines (**A**) K562 and (**B**) HeLa and the resulting proximal RBP interaction intensity. The grey values represent null values within the heatmaps. (**C**) An illustration of the proposed RBP sponge model, portraying lncRNA–RBP proposed behavior, in the presence and absence of a lncRNA. C(I) demonstrates when the lncRNA is present, and C(II) demonstrates the proliferated RBP interactions with mRNAs when the corresponding lncRNA is absent or less abundant. (**D**) Shows the extent of RBPs interacting with lncRNA LINC00909 (data from ENCODE), and supports the lncRNA–RBP sponge model.

**Table 1 genes-10-00593-t001:** Ensembl annotations of the list of long non-coding RNAs (lncRNAs) and corresponding cell lines in which wildtype and knockout RNA-sequencing data were employed in this study.

lncRNA Symbol	ENSG ID	Cell line/s Perturbed In	Length (bp)	Strand	Gene Type
AC016831.7	ENSG00000285106.2	U87	318,897	+	LncRNA
CCAT1	ENSG00000247844.1	HeLa	11,222	-	LincRNA
CTC-428G20.6	ENSG00000271797.1	U87, HeLa	943	+	Antisense
EPB41L4A-AS1	ENSG00000224032.7	K562	4292	+	LncRNA
GAL4	ENSG00000282992.2	HeLa	11,263	-	Protein_coding
KB-1471A8.1	ENSG00000245330.1	HeLa	7069	-	LincRNA
LAMTOR5-AS1	ENSG00000224699.9	HeLa	96,701	+	LncRNA
LINC00263	ENSG00000235823.3	K562, U87, HeLa	80,944	+	LncRNA
LINC00680	ENSG00000215190.9	K562, U87, HeLa	15,427	-	Transcribed_unprocessed_pseudogene
LINC00909	ENSG00000264247.2	K562, U87, HeLa	8568	-	LncRNA
LINC00910	ENSG00000188825.14	K562	50,458	-	LncRNA
MIR142	ENSG00000284353.1	K562	86	-	miRNA
MIR17HG	ENSG00000215417.13	HeLa	6759	+	LncRNA
MIR210HG	ENSG00000247095.3	U87	2797	-	LncRNA
MIR29A	ENSG00000284032.1	U87	63	-	miRNA
PVT1	ENSG00000249859.11	U87, HeLa	392,575	+	LncRNA
RP11-1094M14.11	ENSG00000267321.3	U87, HeLa	6824	+	LncRNA
RP11-126L15.4	ENSG00000236305.1	U87	15,302	-	LncRNA
RP11-347C12.10	ENSG00000260219.1	K562	1086	+	LincRNA
RP11-392P7.6	ENSG00000247498.10	K562	56,919	+	LncRNA
RP11-734K2.4	ENSG00000270344.3	U87	23,411	+	LncRNA
RP11-96L14.7	ENSG00000236782.1	U87	1874	-	Antisense
RP11-973D8.4	ENSG00000258554.1	U87	636	-	Antisense
RP5-1148A21.3	ENSG00000266680.1	HeLa	1403	-	Antisense
SNHG1	ENSG00000255717.7	U87	3969	-	LncRNA
SNHG12	ENSG00000197989.14	U87	4594	-	LncRNA
TRAM2-AS1	ENSG00000225791.7	U87	66,271	+	LncRNA
XLOC_010347	ENSG00000254430.1	U87	699	-	Unprocessed pseudogene
XLOC_014806	ENSG00000211813.2	U87	607	+	TR V gene
XLOC_015111	ENSG00000258687.1	U87	17,537	-	Antisense
XLOC_017263	ENSG00000259656.1	HeLa	1473	+	LincRNA
XLOC_018520	ENSG00000243007.1	K562	495	-	Processed pseudogene
XLOC_026118	ENSG00000272396.1	U87	8285	-	Antisense
XLOC_029037	ENSG00000200718.1	U87	319	+	LncRNA
XLOC_038702	ENSG00000261519.2	K562	1909	+	LincRNA
XLOC_040566	ENSG00000109686.18	U87	222,881	-	Protein_coding
XLOC_042889	ENSG00000271862.1	K562	1588	-	LincRNA
XLOC_051509	ENSG00000253372.5	U87	8771	+	LncRNA
XLOC_054068	ENSG00000268674.1	U87	509	+	Protein_coding

**Table 2 genes-10-00593-t002:** Representation of Fisher’s exact test input contingency table for each iteration of RBP binding data. Conservative hypergeometric tests were performed in order to evaluate the importance of reported proximal RBP binding activity to alternatively spliced genes due to lncRNA knock downs. Each iteration of Fisher’s exact test required this input space, which evaluates if a proximal binding with RBP is occurring in significant frequencies or by chance.

Genetic Regions	Bound	Unbound
Regions where an RBP may bind across genes in a cell line	X	Y
All proximal splicing regions generated from a lncRNA knockdown in a cell	A	B

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
