# Peer review of "Long Non-Coding RNA Expression Levels Modulate Cell-Type-Specific Splicing Patterns by Altering Their Interaction Landscape with RNA-Binding Proteins"

_genes, 2019, doi:10.3390/genes10080593_

Round 1

Reviewer 1 Report

In this work, the authors used publicly available datasets to assess the impact of different lncRNAs knockout on AS, and then they cross their results with the RBP-lncRNA interactome revealed by independent RNA-IP approaches. The work is well written, it is easy to follow and offers a fresh view on the link between lncRNAs and AS in different mammalian cells lines. First, I should say that I find it very interesting how the authors achieved to perform an innovative study using previously published datasets. I consider that this is something we should encourage. Said this, I'd like to suggest a few points prior to publication. 1. We have recently published a review article (Romero-Barrios et al., 2018, NAR) in which we compiled the described lncRNAs involved in AS. In our work we classified the lncRNAs by their action on AS (epigenetics, RNA-RNA duplex, interaction with splicing factors, etc). Based on this, I suggest to take a look if the analyzed lncRNAs are all intergenic or if there is ant antisense or intronic transcript. also, if any neighboring gene of one of these lncRNAs suffers AS in knockout cells. If so, would we consider that there is any epigenetically-mediated mechanism for any particular lncRNA? 2. In plants we can use IsoSwitch (I don't know if it's only useful for humans) as a tool to predict what protein domains would be included or deleted by AS isoforms derived in this case from the knockout of each lncRNA. It'd be very interesting to carry out this analysis for a subset of important AS targets to check if the corresponding encoded proteins will lack or include particular functional domains. 3. I fully understand that this work is focues on human cells, but I suggest the authors to take a look at one of our latest works on a plant lncRNA-interacting splicing factor RNA-IP, and the conclusions about multiple lncRNAs associated to SFactors (Bazin et al., 2018 FPS). One last comment: the quality of the figures is not very high. I found it difficult to read the information on some of them. Perhaps it's due to pdf conversion. Otherwise, please try to improve the quality or simplify the figures to make them easier to read.

Author Response

Review 1

In this work, the authors used publicly available datasets to assess the impact of different lncRNAs knockout on AS, and then they cross their results with the RBP-lncRNA interactome revealed by independent RNA-IP approaches. The work is well written, it is easy to follow and offers a fresh view on the link between lncRNAs and AS in different mammalian cells lines. First, I should say that I find it very interesting how the authors achieved to perform an innovative study using previously published datasets. I consider that this is something we should encourage.

Response: We thank the reviewer for this appreciation of our report. To our knowledge, this is among the first genome-scale analyses which links the interaction landscape of lncRNAs with RNA-binding proteins to rewire post-transcriptional regulatory networks in disease contexts.

Said this, I'd like to suggest a few points prior to publication. 1. We have recently published a review article (Romero-Barrios et al., 2018, NAR) in which we compiled the described lncRNAs involved in AS. In our work we classified the lncRNAs by their action on AS (epigenetics, RNA-RNA duplex, interaction with splicing factors, etc). Based on this, I suggest to take a look if the analyzed lncRNAs are all intergenic or if there is ant antisense or intronic transcript. also, if any neighboring gene of one of these lncRNAs suffers AS in knockout cells. If so, would we consider that there is any epigenetically-mediated mechanism for any particular lncRNA?

Response: We appreciate the feedback from the reviewer.  We now include a Table 1 in the manuscript summarizing the genomic location, cell line, strand and gene type for each of the lncRNAs studied here and have cited review article in introduction of the manuscript. Overall, we found that only six of the lncRNAs studied are anti-sense RNAs according to Ensembl annotations. Although the notion of neighboring gene can be arbitrary proximity, we didn’t find significant evidence for the immediate neighboring genes to be enriched for splicing differences. However, we agree with the reviewer that as more datasets on knockouts for lncRNAs become available, some of these functional questions would be apt in years to come to dissect the various mechanisms through which alternative splicing is modulated by lncRNAs.

In plants we can use IsoSwitch (I don't know if it's only useful for humans) as a tool to predict what protein domains would be included or deleted by AS isoforms derived in this case from the knockout of each lncRNA. It'd be very interesting to carry out this analysis for a subset of important AS targets to check if the corresponding encoded proteins will lack or include particular functional domains.

Response: We again appreciate this very exciting suggestion from the reviewer and agree that such analysis would benefit target selection for further focused studies. However, we feel that current short read data (as opposed to long read transcriptomes which are becoming available) may not accurately reflect the protein domain variations especially for genes which exhibit complex and multiple isoforms. Also, transcript level variants may not be reflected/detected in the proteomic isoforms so evaluating the presence/absence of domains at the proteomic level can be very challenging unless the target is highly expressed.

     However, while our attempts at running IsoSwitch on our human datasets didn’t work, we have attempted to study this phenomenon using Linc00909 as an example. To identify and understand the impacted protein domains by alternative splicing induced by lncRNA perturbation, we used RNA-seq data for Linc00909 knockdown and control in K562 cell line. We performed differential expression analysis using cufflinks-cuffdiff pipeline, to identify differentially expressed transcripts. We also extracted the skipped exon events from our previous analysis for the same Linc00909 in K562 cell line. We then identified which AS skipped exon events were overlapping with differentially expressed (absolute(logFC)>1, p<0.05) transcripts. The genomic locations of exons that were alternatively spliced and differentially expressed were extracted and annotated to protein domain information extracted from Ensembl. We observed that a total of 196 transcripts were differentially expressed in Linc00909 knockdown and control comparison, of these 3 exons were completely overlapping with 3 SE locations. These locations were then mapped to 3 unique genes which included SAC3D1, DMKN and GAREM2 so we anticipate future work to uncover loss or gain of protein function via differential inclusion of functional domains.

I fully understand that this work is focues on human cells, but I suggest the authors to take a look at one of our latest works on a plant lncRNA-interacting splicing factor RNA-IP, and the conclusions about multiple lncRNAs associated to SFactors (Bazin et al., 2018 FPS).

Response: We thank the reviewer for this suggestion and have now cited this work in the introduction of the manuscript to highlight the role of lncRNAs in post-transcriptional control.

One last comment: the quality of the figures is not very high. I found it difficult to read the information on some of them. Perhaps it's due to pdf conversion. Otherwise, please try to improve the quality or simplify the figures to make them easier to read.

Response: The figures included in the integrated manuscript were of low resolution to enhance the download speed for reviewing however, we now upload high quality figures needed for the proof generation of the manuscript.

Reviewer 2 Report

The article by Porto et al  has attempted to analyze impact of lncRNA modulation on post-transcription gene regulation, maily splicing. They have used previously published data from Liu et al (Science 2017) to carryout the analyses. While the concept is new, the reader will benefit from clarity in reason and description in the current article.

1.     The Liu et al have used 6 cancer cell lines and one human iPSCs for CRISPRi targeting >16000 lncRNA loci. Porto et al do not clarify why they chose to analyze data for knockdown of 39 lncRNAs and 3 cell lines. None of the lncRNAs considered in this study are common to all three cell lines. Would it have been more informative to compare effects of a lncRNA knockdown across multiple cell lines? Is more data available to be analyzed beyond the 39lncRNAs and 3 cell lines?

2.      The analyses of cell type specific splicing events (even without lncRNA knockdown) are distracting (Fig. 2 and 3). The splicing events are dissimilar and not a single gene displayed the same patterns. Therefore, Fig. 2 and further analyses of GO functional enrichment are only incremental and distracting from the main focus of the study.

3.     A lncRNA bind multiple RNAs binding protein (RBP). Similarly a RBP can bind multiple lncRNAs. If the lncRNAs are sponging RBPs and have an effect on splicing of the RBP target genes, do the downregulation of two distinct lncRNAs sharing RBP binding sites show similar splicing events  in a given cell line? This may provide another layer of evidence for the hypothesis.

4.     It will be important to show experimental evidence of RBP-sponging by lncRNAs and the downstream effect on splicing of RBP-targets for at least one of the strong predictions eg. Linc00909.

Author Response

Review 2

The article by Porto et al  has attempted to analyze impact of lncRNA modulation on post-transcription gene regulation, maily splicing. They have used previously published data from Liu et al (Science 2017) to carryout the analyses. While the concept is new, the reader will benefit from clarity in reason and description in the current article.

Response: We thank the reviewer for this appreciation of our report. We have now attempted to address the comments from the reviewers in this revised version. We hope the revised version addresses the concerns adequately.

The Liu et al have used 6 cancer cell lines and one human iPSCs for CRISPRi targeting >16000 lncRNA loci. Porto et al do not clarify why they chose to analyze data for knockdown of 39 lncRNAs and 3 cell lines. None of the lncRNAs considered in this study are common to all three cell lines. Would it have been more informative to compare effects of a lncRNA knockdown across multiple cell lines? Is more data available to be analyzed beyond the 39lncRNAs and 3 cell lines?

Response: Liu et. al have performed CRISPR knock outs followed by illumina based short read RNA-sequencing for only 39 lncRNAs in three cell lines and hence our analysis was limited to these datasets. Although the original study attempted to knock out most of the lncRNAs as part of CRINCL library, RNA-sequencing was performed on a subset of these 39  lncRNAs which exhibited potentially interesting phenotypes to follow up with downstream RNA-sequencing. Consequently, our study only used this dataset for splicing analysis. Please note that we have a total of 12 lncRNAs which had knock out RNA-seq in all the three cell lines. However, alternatively spliced genes were not common among the cell lines, for any lncRNA studied here. So we do indeed compare the effects of lncRNA knockdowns across multiple cell lines in our study.

The analyses of cell type specific splicing events (even without lncRNA knockdown) are distracting (Fig. 2 and 3). The splicing events are dissimilar and not a single gene displayed the same patterns. Therefore, Fig. 2 and further analyses of GO functional enrichment are only incremental and distracting from the main focus of the study.

Response: While we agree that further analysis in Figure 3 showing functional enrichment is an extension of the splicing events heatmap shown in Figure 2, we still feel the current narrative helps the reader to better follow the flow of the study by providing a context on how cell type specific splicing events contributed by lncRNAs are also reflected in terms of the functional categories.

A lncRNA bind multiple RNAs binding protein (RBP). Similarly a RBP can bind multiple lncRNAs. If the lncRNAs are sponging RBPs and have an effect on splicing of the RBP target genes, do the downregulation of two distinct lncRNAs sharing RBP binding sites show similar splicing events  in a given cell line? This may provide another layer of evidence for the hypothesis.

Response: We thank the reviewer for this very exciting proposition. Firstly, most lncRNAs interact with multiple RBPs and not necessarily a single RBP. So this is difficult to test. More importantly, our knowledge on the interaction landscape of the RBPs with lncRNAs is far from complete. However, to test this hypothesis one has to have RNA-sequencing data with simultaneous knock out of two different lncRNAs which are both binding to one or more RBPs. Since such data is currently not available for any cell line, it is not possible to test such a hypothesis. Nevertheless, we agree with the reviewer that this can be another layer of regulation which is possible and future integrative approaches can test and validate such models.

It will be important to show experimental evidence of RBP-sponging by lncRNAs and the downstream effect on splicing of RBP-targets for at least one of the strong predictions eg. Linc00909.

Response: While we agree with the reviewer that it would be interesting to experimentally test the hypothesis, we believe such a detailed experimental project is beyond the scope of the current study. We hope to carry out such a study in the next few months where we study the knockout of Linc00909 by performing CLIP-seq of various prominent splicing factors which are extensively interacting with Linc00909 in one of more cell lines.

Round 2

Reviewer 2 Report

Table 1 lists lncRNAs knockdown included in this study. However 3 of these are protein-coding RNAs?

Although this is the first step to analyze complex interaction between lncRNAs and RBPs that may influence alternative splicing events, no two cell lines show a similar effect of lncRNA knockdown and it is not tested if two distinct lncRNAs binding to a RBP associated with similar alternative splicing events upon lncRNA knockdown. The authors mention in their response that 'to test this hypothesis one has to have RNA-sequencing data with a simultaneous knock out of two different lncRNAs which are both binding to one or more RBPs. Since such data is currently not available for any cell line, it is not possible to test such a hypothesis'

The dataset the authors describe in this manuscript does include knockdown of two lncRNAs in a single cell line with a similar binding intensity to RBPs eg. LINC00909 and LINC00910 both are predicted to bind YBX3 with very high intensity in K562 cell line (Fig. 4A).

Author Response

Reviewer 2

Table 1 lists lncRNAs knockdown included in this study. However, 3 of these are protein-coding RNAs?

Response: We appreciate the observation from the reviewer regarding this aspect in Table 1. However, please note that several lncrnas can behave as protein coding RNAs and this can be evidenced from several studies listed below. Hence, in our list of studied lncRNAs some of them are currently also known to be protein coding transcripts. For instance, transcript of GAL4 gene depending on the presence of glucose can behave as a coding or non-coding RNA.

https://academic.oup.com/bib/advance-article/doi/10.1093/bib/bby055/5047384

https://www.ncbi.nlm.nih.gov/pubmed/16670683

https://www.ncbi.nlm.nih.gov/pubmed/8196626

https://www.ncbi.nlm.nih.gov/pubmed/27700226

Although this is the first step to analyze complex interaction between lncRNAs and RBPs that may influence alternative splicing events, no two cell lines show a similar effect of lncRNA knockdown and it is not tested if two distinct lncRNAs binding to a RBP associated with similar alternative splicing events upon lncRNA knockdown. The authors mention in their response that 'to test this hypothesis one has to have RNA-sequencing data with a simultaneous knock out of two different lncRNAs which are both binding to one or more RBPs. Since such data is currently not available for any cell line, it is not possible to test such a hypothesis'

The dataset the authors describe in this manuscript does include knockdown of two lncRNAs in a single cell line with a similar binding intensity to RBPs eg. LINC00909 and LINC00910 both are predicted to bind YBX3 with very high intensity in K562 cell line (Fig. 4A).

Response: We appreciate the suggestion from the reviewer. We have now examined the case of LINC00909 and LINC00910 both of which interact with at least 50 RBPs each across cell lines for which we have the data from ENCODE’s eCLIP sequencing. In particular, LINC00909 interacts with 63 RBPs and LINC00910 interacts with 79 RBPs across cell lines. Out of which 49 are common RBPs. Thus, YBX3 may not be the only contributor to the final outcome of splicing in a cell type in this case. Comparing the splicing signatures from the heatmap in Figure 2 indicates that there is very low overlap in the set of genes which are spliced suggesting that although they share the interacting RBPs, our data doesn’t seem to suggest significant overlap in the splicing outcome profile. However, we agree that this is too small sample size to make any concluding statement about the overlap of splicing profiles, when the lncRNAs share interacting RBPs. We anticipate future datasets and studies can provide more conclusive interpretation of the dynamics between lncRNAs and RBPs to modulate splicing. Also, our notion of knock out of two lncRNAs in the previous response corresponded to double knock out of the lncRNAs in a given cell line to understand the impact of multiple lncRNAs. However, such double knock outs of lncRNAs are currently not available.